# Vitamin D levels of pregnant immigrant women and developmental disorders of language, learning and coordination in offspring

**Bianca Arrhenius**[1,2], **Heljä-Marja Surcel**[3,4], **Susanna Hinkka Yli-Salomäki**[1,2], **Alan Brown**[5,6], **Keely Cheslack-Postava**[5], **Andre Sourander**[1,2,7] *

1 Department of Child Psychiatry, University of Turku, Turku, Finland, 2 INVEST Research Flagship, University of Turku, Turku, Finland, 3 Faculty of Medicine, University of Oulu, Oulu, Finland, 4 Biobank Borealis of Northern Finland, Oulu University Hospital, Oulu, Finland, 5 Department of Psychiatry, New York State Psychiatric Institute, Columbia University, New York, New York, United States of America, 6 Department of Epidemiology, Columbia University Mailman School of Public Health, New York, New York, United States of America, 7 Department of Child Psychiatry, Turku University Hospital, Turku, Finland

* andsou@utu.fi

**Data Availability Statement:** Individual level data cannot be shared due to Finnish data protection law on the use of national register data. Our study

## Abstract

### Background

Prenatal vitamin D deficiency is a common health concern among immigrants. No previous studies have examined the associations between prenatal vitamin D levels and developmental disorders of language, scholastic skills, and coordination in an immigrant sample.

### Methods

The sample included 542 immigrant mothers of cases with language, scholastic, coordination or mixed developmental disorders, 443 immigrant mothers of controls without these disorders and 542 Finnish mothers of controls. Maternal vitamin D was measured in serum samples collected during early pregnancy and stored in a national biobank.

### Results

The mean vitamin D levels during pregnancy were 25.0 (SD 14.4) nmol/L for immigrant mothers of cases, 25.4 (SD 15.5) for immigrant mothers of controls and 42.3 (SD 19.1) for Finnish mothers of controls. Low maternal vitamin D levels during pregnancy were not associated with the selected developmental disorders in offspring when immigrant mothers of cases were compared to immigrant mothers of controls (adjusted OR for continuous log-transformed vitamin D: 1.01, 95% CI 0.75–1.36, p = 0.96). When immigrant mothers of cases were compared to Finnish mothers of controls, the adjusted OR for continuous vitamin D was 18.94 (95% CI 11.47–31.25), p <0.001. The results were similar when vitamin D was examined as a categorical variable or divided into quintiles.

approvals from the National Institute of Health and Welfare and Biobank Borealis, from which the data were obtained, prohibit sharing any individual level data. Analytic code and data dictionary of vitamin D data is available at Biobank Borealis (https://oys.fi/biopankki/briefly-in-english/), and data dictionary of the register data of is available at the Finnish Institute of Health and Welfare (https://thl.fi/en/statistics-and-data/data-and-services/register-descriptions/newborns for FMBR and https://thl.fi/en/statistics-and-data/data-and-services/register-descriptions/care-register-for-health-care for CRHC). Any researcher may repeat our study process and compile his/her own study data base. Information on the process to request data from Finnish Institute of Health and Welfare is available at https://findata.fi/en/permits/data-requests/ and for biological samples in Biobank Borealis at https://oys.fi/biopankki/for-researchers/

**Funding:** This research was supported supported by the INVEST Research Flagship Centre/Research Council of Finland (decision number: 345546).

**Competing interests:** The authors have declared that no competing interests exist.

## Conclusions

Prenatal vitamin D levels were low, and similar, among immigrant mothers of cases with selected developmental disorders and unaffected controls. This indicates that vitamin D unlikely mediates previously observed associations between maternal immigrant status and the selected developmental disorders in offspring. The proportion of immigrant mothers with severe vitamin D deficiency was very high, which underlines the importance of prenatal counselling and overall public health efforts to improve immigrant health.

## Introduction

Neurodevelopmental disorders and autism spectrum disorder in particular have been found more prevalent among children of immigrants than nationals in several studies [1, 2]. In a Finnish sample, developmental disorders of speech and language, scholastic skills and coordination were also more common among children of immigrants [3]. One hypothesized explanation for these associations has been vitamin D deficiency during pregnancy, a condition that is common among immigrants especially in the Nordic countries, where sunlight is scarce from late Autumn to early Spring [4, 5].

Vitamin D is a fat-soluble vitamin, with the main function in the body of regulating two essential nutrients, namely, calcium and phosphate. Vitamin D also affects brain development, and findings from both human and animal studies have indicated possible associations between low vitamin D levels during pregnancy and different developmental outcomes in offspring [6]. Prenatal vitamin D deficiency during pregnancy has been associated with autism and attention deficit hyperactivity disorder (ADHD) in offspring [7, 8], whereas the associations for cognitive impairments and other neurodevelopmental outcomes have been inconclusive [6, 9]. A meta-analysis indicated possible small adverse effects of prenatal vitamin D deficiency on cognitive and language development [10]. In our previous study including over 3,000 pregnant women, vitamin D deficiency in early pregnancy was not associated with specific learning disorders in offspring in a Finnish sample [11]. However, some studies have indicated possible effect moderation by race or ethnic group on the associations between prenatal vitamin D levels and developmental outcomes [12, 13]. Reviews have called for more research of possible effects of ethnicity on vitamin D deficiency and neurodevelopmental outcomes, as most studies have been conducted on Caucasian populations.

The previously observed lower prenatal vitamin D levels among pregnant immigrant women and the higher prevalences of developmental disorders among immigrant children raises the question whether vitamin D might mediate associations between maternal immigration status and developmental disorders in offspring. This nationwide biomarker study aimed to find out whether the vitamin D levels of pregnant immigrant mothers are associated with subsequent risk of their child being diagnosed with developmental disorders of speech and language, scholastic skills or coordination. For this purpose, the vitamin D levels of immigrant mothers whose offspring were diagnosed with the selected developmental disorders were compared to the vitamin D levels of immigrant mothers of unaffected controls. We hypothesized that there would be an association between vitamin D deficiency among immigrant mothers and developmental disorders of speech and language, scholastic skills or coordination in offspring. We also wished to evaluate the relationship between immigrant status and vitamin D

levels during pregnancy. For this purpose, we compared vitamin D levels of immigrant mothers of cases with maternal vitamin D levels of native Finnish mothers of unaffected controls.

## Materials and methods

This study is part of a nationwide register-based nested case-control study that includes all singleton live births in Finland from 1.1.1996 to 31.12.2007, with diagnoses available from the Care Register for Health Care (CRHC) by 31.12.2012. The ethical approval for the study was provided by the Ethics Committee of the Wellbeing County of Southwest Finland. Additionally, the data protection authorities at the Finnish Institute for Health and Welfare provided access to the register data and the Biobank Borealis in Oulu provided access to the blood samples stored in the national biobank. The data was handled and pseudonymized according to Finnish data protection laws. No cases or controls were contacted, and therefore, informed consent was not required for the use of register data. The pregnant women provided informed written consent to use their prenatal routine screening serum samples from maternity clinics for research purposes. The biobank data was accessed 31.12.2017.

### Finnish Maternity Cohort

The Finnish Maternity Cohort (FMC) includes two million maternal serum samples, which were collected in the first trimesters of pregnancy (majority of the samples in weeks 8–16 of gestation) from almost one million women. The samples are stored at −25˚C in a protected biorepository at Biobank Borealis in Oulu, Finland, and available for scientific research. The unique personal identification code, which is assigned to all Finnish residents, was used to link the FMC samples with other Finnish registers.

### National registers

The Care Register for Health Care (CRHC) is a national register for diagnoses made during inpatient and outpatient visits to Finnish public health care, and is maintained by the Finnish Institute for Health and Welfare (THL). The CRHC was used to identify the cases, which were selected as those with registered diagnoses of speech or language, scholastic, coordination, or mixed developmental disorders by the end of 2012 among children to immigrant mothers. The CRHC also provided information on other diagnoses among cases and controls and parental psychopathology. The International Classification of Diseases (ICD) classification is used to code the diagnoses in the register.

The Finnish Maternity Birth Register (FMBR) was used to extract variables related to maternal health and pregnancy, delivery and newborn health. The Digital and Population Data Services Agency (DVV) was used to identify the controls and to obtain information on the subjects' parents. The DVV manages the demographic information of everyone living in Finland, which includes the name, personal identification code, address, native language, country of birth, citizenship, family information and date of birth.

### Cases and controls

The cases were born between 1996 and 2006 in Finland and diagnosed with a speech or language disorder (ICD-10 F80), scholastic disorder (F81), coordination disorder (F82), or mixed developmental disorder (F83) by 31.12.2012. Cases with comorbid autism spectrum disorder (ASD, F84) and/or intellectual disability (ID, F70–79) were excluded because these diagnoses conflict with the definition of the included outcome diagnoses.

Cases were restricted to offspring of mothers who were immigrants. Maternal immigrant status was defined as being born in Africa, the Middle East, Asia or South and Latin America in countries with low or medium human development index (HDI). This definition was chosen to minimize the number of Western/Caucasian immigrants in the sample. The HDI index is a global categorization that reflects the education and income level as well as the life expectancy in different countries. The scores for the three HDI dimension indices (education, income and life expectancy) are then aggregated into a composite index using the geometric mean and countries can be defined as having very high ($\geq 0.800$), high (0.700–0.799), medium (0.550–0.699) or low human development ($< 0.550$). The Human Development report from 2018 was used to categorize the countries [14].

Disorders of speech and language are characterized by disturbances in normal patterns of language acquisition. Scholastic disorders, also known as specific learning disorders, refer to impairments in reading, spelling or arithmetical skills. Coordination disorders are characterized by impaired fine and gross motor coordination development, without previous traumatic accidents or surgery. Mixed developmental disorders cover a combination of developmental disorders of speech and language, scholastic skills and motor function, but where no disorder is dominant enough for a primary diagnosis. The analyses in this study were conducted for the group of children with any of these developmental disorders, i.e., one or more of the included diagnoses (ICD-10: F80–F83). In Finland, developmental disorders are usually diagnosed in publicly funded outpatient clinics of pediatric neurology, pediatrics, phoniatrics or child psychiatry. The procedures for diagnosis include standardized tests and have been described previously [15].

The cases were matched 1:1 both to controls with immigrant mothers and controls with Finnish mothers. In both control groups, the offspring could not have a diagnosis of speech and language, scholastic, coordination or mixed developmental disorders, ASD, or ID and had to be singletons, alive and resident in Finland when the matched case was diagnosed. Controls were selected by linking the CRHC and the DVV registers.

Finnish mothers were defined as those either born in Finland or native speakers of one of Finland's three official languages: Finnish, Swedish or Sami. Controls with Finnish mothers were matched with the cases by sex and date of birth, plus or minus 30 days. Controls with immigrant mothers were available in limited numbers, therefore the controls with immigrant mothers were allowed to be of different sex than the case and required to be born within 5 years of the case. We aimed to match the immigrant mothers of cases and controls by their birth country, but if there was no control available from the same country, we used the following criteria: the matched control's mother should be born in a country from the same continent and with the same HDI as the case mother's birth country.

## Maternal serum vitamin D evaluation

The measurement of maternal serum vitamin D (25(OH)D) was performed blind to case-control status using a chemiluminescence microparticle immunoassay by an Architect i2000SR automatic analyzer (Abbott Diagnostics). The procedure has been described previously [8]. The blood samples were collected both before and after (11.5.1995–10.11.2006) the national food fortification with vitamin D and the supplementation recommendations of 10 micrograms of vitamin D per day for pregnant women started in Finland in 2003.

## Covariates

Potential confounding factors that have been associated with maternal vitamin D levels, immigrant status and the selected developmental disorders in previous studies were initially selected

for covariate testing, i.e., testing for associations with both exposure and outcome. Information on maternal smoking, age, number of previous births, gestational diabetes and socioeconomic status, as well as offspring gestational age, birth weight and Apgar score, were obtained from the FMBR and the season and gestational week of blood draw from the FMC. Data on maternal and paternal psychopathology and history of maternal substance abuse were obtained from the CRHC. For the classification of the covariates, see S1 Tables 1 and 2 in S1 File.

## Statistical analysis

Maternal vitamin D was examined as a continuous variable, which was natural log transformed before the analyses due to a skewed distribution. Maternal vitamin D levels were also categorized into quintiles and clinical categories. The cut-off points for the quintiles were based on the distribution of maternal vitamin D in the corresponding control group. The following clinical categories were used: (1) deficient (25(OH)D <30 nmol/L), (2) insufficient (25 (OH)D 30–49.9 nmol/L), and (3) sufficient (25(OH)D >50 nmol/L). The highest quintile and the sufficient category served as reference groups.

In the comparison of immigrant cases vs. immigrant controls, the associations between the covariates and 1) the exposure, i.e., maternal 25(OH)D levels among immigrant mothers of controls and 2) the outcome, i.e., the selected developmental disorders, were tested with Student's T-tests, ANOVA (categorical covariate variables) or linear regression (continuous covariate variables) and conditional logistic regression for the association with the outcome. The same procedure for covariate testing was performed in the comparison of immigrant mothers of cases vs. Finnish mothers of controls. Covariates were included in the adjusted model if they were associated with both exposure and outcome at p < 0.1. We used conditional logistic regression for matched pairs to calculate odds ratios (ORs) with 95% confidence intervals (CIs) for the associations between prenatal vitamin D levels and the outcome diagnoses. Sex interaction was tested using logistic regression with a product term for the sex X exposure variable. Subgroup analyses for comorbid ADHD were performed with continuous vitamin D as the exposure variable. Groups were defined as cases with comorbid ADHD (ICD-10 F90) or without ADHD. Statistical significance was based on p < 0.05 in all other analyses except for the covariate testing. SAS statistical software was used to perform the analyses (SAS 9.4, SAS Institute, Cary, NC, USA).

## Results

Among all 612,315 singleton-born children in Finland between 1.1.1996 and 31.12.2006, 791 children with immigrant mothers from low and medium HDI countries in Africa, the Middle East, Asia or South and Latin America were diagnosed with a language, scholastic, coordination or mixed developmental disorder in specialized health care by the end of 2012. Of these children, 88 (11.1%) had ID, 24 (3.0%) had ASD, and 20 (2.5%) had both ID and ASD and were therefore excluded. Of the remaining 659 cases with immigrant mothers, 542 cases and 542 matched controls with Finnish mothers had a maternal serum sample available in the FMC collection. Due to limited availability of controls with immigrant mothers, 443 cases were matched with 443 controls with immigrant mothers. These 443 cases did not differ from the cases who could not be matched to an immigrant control by sex, diagnosed outcome disorder or mother's birth continent (S1 Table).

Most of the immigrant mothers came from Asia: 277 (51.1%) of the 542 case mothers and 224 (50.6%) of the 443 control mothers. The categorized countries of origin and HDI of the immigrant subjects are presented in Table 1. S1 Appendix provides a list of the included immigrant mothers' birth countries. The mean age at offspring's diagnosis was 5.5 years (SD 1.7) for

**Table 1. Immigrant sample characteristics.**

| Country of origin, categorized | Immigrant mothers to cases N = 542 | Immigrant mothers to controls N = 443 |
|---|---|---|
| Sub Saharan Africa | 224 (41.3) | 187 (42.2) |
| North Africa and Middle East | 34 (6.3) | 27 (6.1) |
| Asia | 277 (51.1) | 224 (50.6) |
| Latin and South America | 7 (1.3) | 5 (1.1) |
| **Human development index of mother's country of origin** | | |
| Low | 217 (40.0) | 184 (41.5) |
| Medium | 325 (60.0) | 259 (58.5) |

language disorder, 7.9 years (SD 2.5) for scholastic disorder, 5.2 years (SD 2.1) for coordination disorder, and 6.1 years (SD 2.2) for mixed developmental disorder. Among the 542 cases and controls matched by sex and birth month, 387 (71.4%) were male and 155 (28.6%) were female. The prevalence of comorbid ADHD among cases was 42 of 542 (7.9%). Among the controls with immigrant mothers, who could not be matched by sex, birth year and season of birth, 295 (66.6%) were male and 148 (33.4%) were female, whereas the corresponding numbers among the 443 cases with immigrant mothers were 314 (70.9%) and 129 (29.1%). Only 13 (2.9%) of the 443 controls with immigrant mothers had a different birth year than their corresponding case. Season of birth did not differ significantly between the cases and controls with immigrant mothers (p-value range 0.29–0.43 for the four seasons).

The mean gestational week of maternal blood collection was 12.7 (SD 5.6) for immigrant mothers of cases, 12.9 (SD 5.4) for immigrant mothers of controls and 10.2 (SD 2.6) for Finnish mothers of controls. The median 25(OH)D level was 25.0 nmol/L (SD 14.4; range 7.7–109.8 nmol/L) for immigrant mothers of cases, 25.4 nmol/L (SD 15.5; range 7.0–95.6nmol/L) for immigrant mothers of controls and 42.3 nmol/L (SD 19.1; range 13.7–155.0 nmol/L) for Finnish mothers of controls. When the mean vitamin D levels were examined by year of blood draw, there were no differences in the vitamin D levels of immigrant mothers of cases after the national food fortification and vitamin D supplementation recommendations started in Finland in 2003, whereas a slight increase was observed among immigrant mothers of controls and a notable increase among mothers of Finnish controls (Fig 1 and S2 Table). Furthermore, the time from immigration to blood draw was not associated with the vitamin D levels of immigrant mothers. In the categorized vitamin D groups, the mean time from immigration was 5.0 (SD 4.1) years for the group with vitamin D < 30nmol/L, 5.0 (SD 4.6) years for those with vitamin D 30–50 nmol/L and 4.5 (SD 3.6) years for those with vitamin D >50 nmol/L (p = 0.5 for linear association).

In the comparison of cases with immigrant mothers vs. controls with immigrant mothers, the analyses were initially adjusted for offspring sex, birth year and season of birth, because the immigrant subjects could not be matched by these characteristics. In addition, previous births and maternal age were significantly associated with both vitamin D levels among immigrant mothers of controls and any diagnosis of the selected developmental disorders (S1 Table 1 in S1 File). In the comparison of immigrant cases vs Finnish controls, the following covariates were found significant in the covariate testing: maternal age, offspring weight for gestational age and Apgar score (S1 Table 2 in S1 File).

In the comparison of cases with immigrant mothers vs. controls with immigrant mothers, no significant associations were found between low maternal vitamin D and the selected developmental disorders when vitamin D was examined as a continuous log-transformed variable

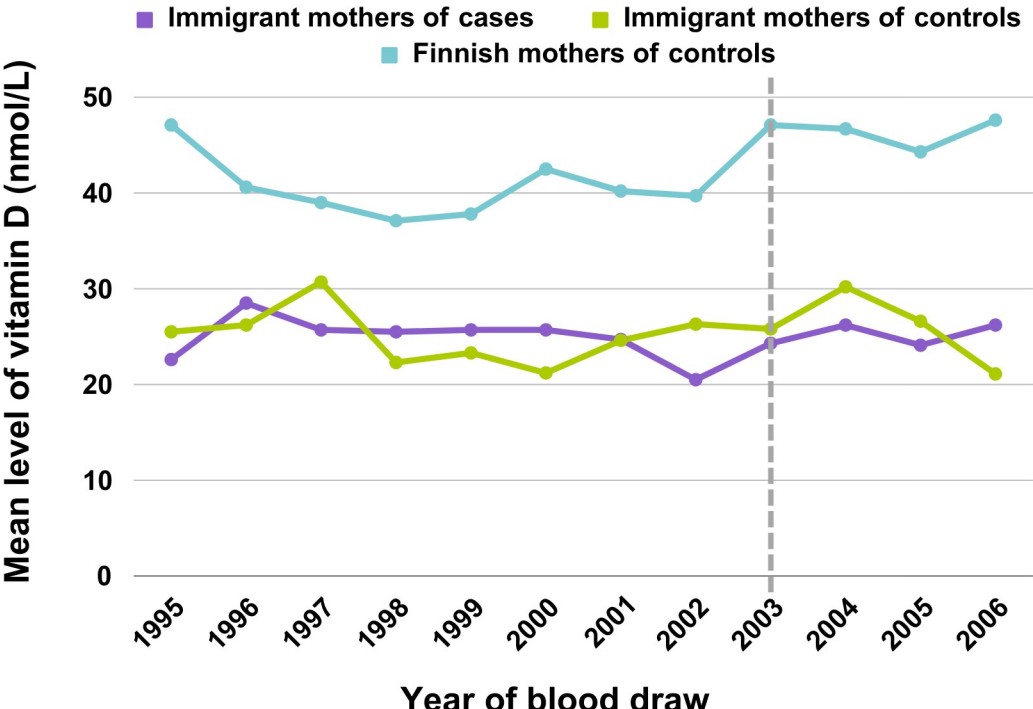

**Fig 1. Changes over time in mean vitamin D levels among immigrant and Finnish pregnant women.** The dashed line marks the start of the increased vitamin D recommendation and food fortification in Finland in 2003. For frequencies and standard deviations, see S2 Table.

(aOR 1.01 (95% CI 0.75–1.36, p = 0.96), categorical variable (25(OH)D <30 nmol/L: aOR 1.21 (95% CI 0.74–2.00, p = 045) or divided into quintiles (aOR for the lowest versus highest quintile 0.93 (95% CI 0.57–1.51), p = 0.76, Table 2 and Fig 2). The results did not differ in the crude and adjusted analyses (Table 2). Comorbid ADHD did not affect the results: the aOR for the group with ADHD was 0.50 (95% CI 0.14–1.77, p = 0.28) and it was 1.05 (95% CI 0.77–1.44, p = 0.76) for the group without ADHD.

In the comparison of cases with immigrant mothers vs. controls with Finnish mothers, there were significant associations between low maternal vitamin D and the selected developmental disorders when vitamin D was examined as a continuous log-transformed variable (aOR 18.94, 95% CI 11.47–31.25, p <0.001), categorical variable (25(OH)D <30 nmol/L: aOR 14.41, 95% CI 8.34–24.90, p <0.001), and divided into quintiles (aOR for the lowest quintile 37.75, 95% CI 17.56–81.15, p <0.001) (Table 2 and Fig 2). The aOR for the group with comorbid ADHD was 57.14 (95% CI 4.67–699.25, p = 0.002) and it was 18.41 (95% CI 10.98–30.87, p<0.001) for the group without ADHD. There was no interaction by sex for the association between continuous log-transformed maternal vitamin D and the selected developmental disorders in either comparison.

## Discussion

This is the first study to examine the association between maternal vitamin D levels in prenatal sera and diagnosed developmental disorders of speech and language, scholastic skills and coordination in a large immigrant sample. We found that maternal vitamin D levels were low, and similar, among immigrant mothers of cases with the selected developmental disorders and immigrant mothers of unaffected controls. This indicates that vitamin D deficiency unlikely

**Table 2. Odds ratios of the association between maternal serum vitamin D during pregnancy and developmental disorders of speech and language, scholastic skills, coordination, or mixed type in offspring.**

| Comparison 1: Immigrant mothers to cases vs immigrant mothers to controls | | | | | | | |
|---|---|---|---|---|---|---|---|
| **Vitamin** | **Immigrant mothers to cases** | **Immigrant mothers to controls** | **OR[1]** | **P -value** | **Adjusted[2]OR** | **P -value** |
| | N = 443 | N = 443 | 95% CI | | 95% CI | |
| | Mean (SD) | Mean (SD) | | | | |
| **Continuous** Log-transformed analysis[3] | 25.1 (14.4) | 25.4 (15.5) | 1.01 (0.76–1.34) | 0.83 | 1.01 (0.75–1.36) | 0.96 |
| **Quintiles** | **n (%)** | **n (%)** | | | | |
| < 20 | 86 (19.4) | 91 (20.5) | 0.99 (0.61–1.59) | 0.95 | 0.93 (0.57–1.51) | 0.76 |
| 20–39 | 83 (18.7) | 89 (20.1) | 0.96 (0.61–1.50) | 0.84 | 0.96 (0.60–1.53) | 0.85 |
| 40–59 | 99 (22.4) | 86 (19.4) | 1.18 (0.77–1.82) | 0.44 | 1.16 (0.75–1.81) | 0.51 |
| 60–79 | 90 (20.3) | 90 (20.3) | 1.01 (0.67–1.52) | 0.97 | 1.03 (0.68–1.57) | 0.89 |
| ≥80 | 85 (19.2) | 87 (19.6) | Reference | | Reference | |
| **Categories** | **n (%)** | **n (%)** | | | | |
| <30 | 330 (74.5) | 330 (74.5) | 1.22 (0.75–1.99) | 0.42 | 1.21 (0.74–2.00) | 0.45 |
| 30 - <50 | 79 (17.8) | 73 (16.5) | 1.37 (0.78–2.40) | 0.27 | 1.38 (0.78–2.44) | 0.27 |
| ≥ 50 | 34 (7.7) | 40 (9.0) | Reference | | Reference | |
| **Comparison 2: Immigrant mothers to cases vs Finnish mothers to controls** | | | | | | | |
| **Vitamin D** | **Immigrant mothers to cases** | **Finnish mothers to controls** | **OR** | **P -value** | **Adjusted[4] OR** | **P -value** |
| | N = 542 | N = 542 | 95% CI | | 95% CI | |
| | Mean (SD) | Mean (SD) | | | | |
| **Continuous** Log-transformed analysis[3] | 25.0 (14.4) | 42.3 (19.1) | 18.53 (11.52–29.79) | <0.001 | 18.94 (11.47–31.25) | <0.001 |
| **Quintiles** | **n (%)** | **n (%)** | | | | |
| < 20 | 364 (67.2) | 110 (20.3) | 32.73 (16.10–66.54) | <0.001 | 37.75 (17.56–81.15) | <0.001 |
| 20–39 | 71 (13.1) | 111 (20.5) | 5.74 (2.80–11.74) | <0.001 | 6.57 (3.06–14.11) | <0.001 |
| 40–59 | 49 (9.0) | 106 (19.6) | 3.76 (1.86–7.59) | <0.001 | 4.49 (2.11–9.56) | <0.001 |
| 60–79 | 38 (7.0) | 107 (19.7) | 2.35 (1.11–4.95) | 0.025 | 2.62 (1.20–5.73) | 0.016 |
| ≥80 | 20 (3.7) | 108 (19.9) | Reference | | Reference | |
| **Categories** | **n (%)** | **n (%)** | | | | |
| <30 | 403 (74.4) | 166 (30.6) | 13.92 (8.30–23.35) | <0.001 | 14.41 (8.34–24.90) | <0.001 |
| 30 - <50 | 101 (18.7) | 222 (41.0) | 2.28 (1.40–3.71) | <0.001 | 2.30 (1.37–3.86) | 0.002 |
| ≥ 50 | 38 (7.0) | 154 (28.4) | Reference | | Reference | |

[1]Adjusted for sex, birth year and season of birth because the groups were not initially matched by these characteristics.

[2]Adjusted for sex, birth year, season of birth, parity and maternal age.

[3] For one nmol/l decrease in vitamin D levels.

[4]Adjusted for apgar score, weight for gestational age and maternal age.

affected the risk of language, learning and coordination disorders in offspring to immigrant mothers. The findings are in line with some previous studies that have not found any associations between maternal serum vitamin D and learning-related outcomes, such as academic achievement, IQ or specific learning disorders [11, 16, 17].

We also found that that immigrant mothers are at high risk for severe vitamin D deficiency during pregnancy compared to Finnish mothers. The mean vitamin D levels among immigrant mothers was ~25 nmol/L for both cases and controls in this study, which is, by most categorical definitions of vitamin D deficiency, alarmingly low. The definition of what is considered as deficiency varies somewhat in the literature, but most previous studies have

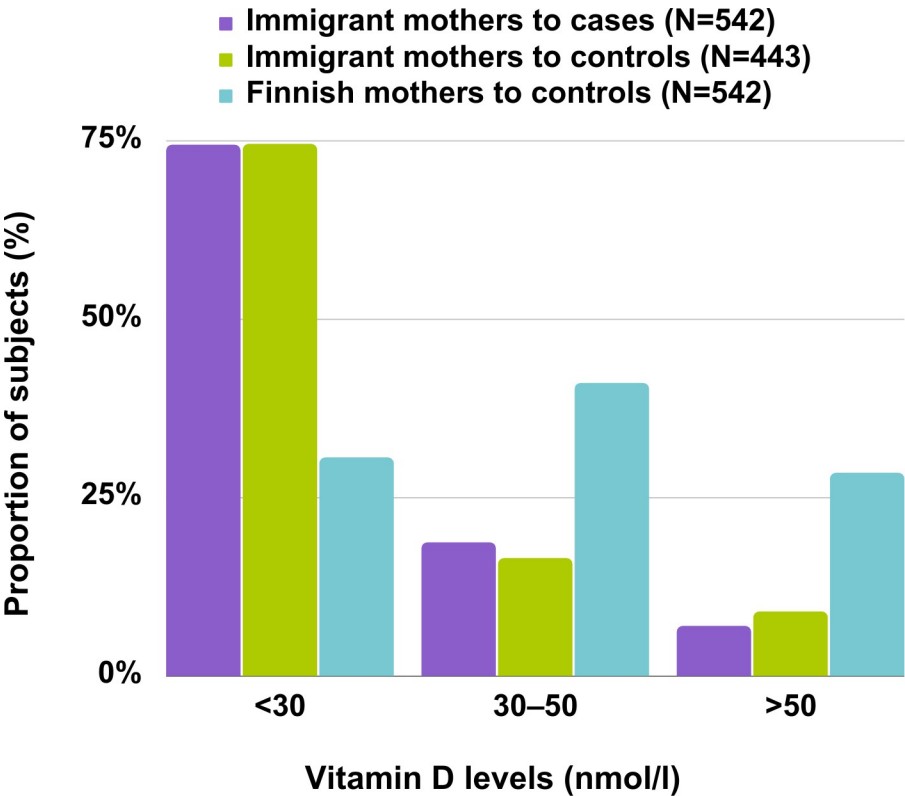

**Fig 2. The proportion of cases and controls in categories of prenatal vitamin D levels.** Both immigrant cases and controls displayed similarly low levels of vitamin D compared to Finnish controls. See Table 2 for frequencies.

used <25–30 or <50 as the definition of vitamin D deficiency [18]. Vitamin D deficiency during pregnancy has been associated with adverse outcomes such as gestational diabetes, pre-eclampsia, and offspring low birth weight [19]. The underlying reasons for vitamin D deficiency among immigrants are potentially related to skin type and the more prevalent use of covering clothes [20], but also lack of knowledge about the sources of vitamin D and supplement recommendations during pregnancy [21]. Data from THL indicates that pregnant women who were not born in Finland used maternity health care services slightly less than women born in Finland. For example, in year 2019, the mean number of maternity clinic visits for Finnish-born women was 14.0 (SD 4.7) and it was 13.1 (SD 4.3) for immigrant women (data source: personal correspondence with THL). Another report by THL also found that immigrants were less satisfied with public health services than Finnish natives [22]. Hence, the impact of preventive efforts including the use of vitamin D supplements might be lower in immigrant populations and this could partly explain why the change in the Finnish vitamin D supplementation recommendations in 2003 [23] had no or little impact on the mean vitamin D levels of immigrant mothers, whereas an increase was observed for Finnish mothers in this study.

Besides vitamin D deficiency, the high prevalence of developmental disorders among children of immigrant parents in Finland is another important issue that has been demonstrated using partially overlapping national data with the current paper [3]. Developmental disorders of speech and language, scholastic skills and coordination are a heterogeneous group, and their etiology is multifactorial, with strong genetic components [24]. However, there is no previous evidence suggesting that immigrants who come to Finland would have higher genetic

risks for such disorders. Lower socioeconomic status and educational level among migrants may play a role [15] and it has been speculated that clinicians might misattribute other social, behavioral or language problems among immigrant children as developmental disorders [25]. Furthermore, prenatal stress due to traumatic experiences might contribute to unfavorable perinatal circumstances and developmental outcomes in offspring [26].

The strengths of this study include a large prospectively collected biobank and register sample of immigrant mothers and their offspring. Some limitations also need to be considered. First, the register-based diagnoses were collected from specialized services only, which limits the conclusions to cases with language, scholastic and coordination disorders who reached these services. However, in Finland, all pregnant women and their children receive free regular health check-ups at publicly funded health clinics, and these services are used by over 99% of the population, including immigrants. It is therefore unlikely that children with apparent developmental difficulties would have been missed and not referred to specialized services due to the lack of health care services. Second, all the immigrant cases could not be matched with immigrant controls due to the limited availability of immigrant controls. However, the cases with and without immigrant controls had similar characteristics, lowering the risk for bias. A third limitation is that the blood samples in the study were collected in the first or early second trimester of the pregnancy, therefore the possible effect of maternal vitamin D deficiency during later phases of pregnancy on the selected developmental disorders in offspring remains unclear. However, previous literature suggests that early pregnancy may be particularly crucial for fetal brain development and the possible adverse effects of vitamin D deficiency [9, 27]. Vitamin D levels also typically remain quite stable throughout pregnancy [28].

## Conclusions

First, we found that prenatal vitamin D levels were not associated with language, learning and coordination disorders in immigrant offspring. Second, we found a very high prevalence of vitamin D deficiency among pregnant immigrant women. The findings have important public health significance, because vitamin D deficiency has been associated with a wide range of medical conditions in both pregnancy and offspring. Because of the increase of immigrant populations globally, the results highlight the need to find culturally sensitive ways to address the health needs of immigrant women.

## Supporting information

**S1 File. Classification of covariates and relationship between covariates, exposure and outcome.**
(DOCX)

**S1 Appendix. Birth countries of immigrant mothers to cases and controls.**
(DOCX)

**S1 Table. Descriptive characteristics of immigrant subjects with and without immigrant controls.**
(DOCX)

**S2 Table. Yearly means of maternal vitamin D in each group of cases and controls.**
(DOCX)

## Acknowledgments

The authors would like to thank Emmi Heinonen for her contribution to the statistical analyses. Further thanks to research personnel at THL for providing data on maternity health clinic service use among immigrants.

## Author Contributions

**Conceptualization:** Heljä-Marja Surcel, Susanna Hinkka Yli-Salomäki, Andre Sourander.

**Data curation:** Susanna Hinkka Yli-Salomäki.

**Formal analysis:** Susanna Hinkka Yli-Salomäki.

**Funding acquisition:** Alan Brown, Andre Sourander.

**Investigation:** Bianca Arrhenius, Heljä-Marja Surcel.

**Methodology:** Bianca Arrhenius, Heljä-Marja Surcel, Susanna Hinkka Yli-Salomäki, Alan Brown, Keely Cheslack-Postava, Andre Sourander.

**Project administration:** Alan Brown, Andre Sourander.

**Resources:** Heljä-Marja Surcel.

**Supervision:** Andre Sourander.

**Validation:** Heljä-Marja Surcel.

**Visualization:** Bianca Arrhenius.

**Writing – original draft:** Bianca Arrhenius.

**Writing – review & editing:** Heljä-Marja Surcel, Susanna Hinkka Yli-Salomäki, Alan Brown, Keely Cheslack-Postava, Andre Sourander.

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
