## [Decision Letter · Decision Letter 0]

5 Dec 2023

PONE-D-23-33791Prenatal vitamin D levels of immigrant mothers and language, scholastic, coordination and mixed developmental disorders in offspringPLOS ONE

Dear Dr. Arrhenius,

Thank you for submitting your manuscript to PLOS ONE. After careful consideration, we feel that it has merit but does not fully meet PLOS ONE’s publication criteria as it currently stands. Therefore, we invite you to submit a revised version of the manuscript that addresses the points raised during the review process.

We look forward to receiving your revised manuscript.

Kind regards,

Ramune Jacobsen

Academic Editor

PLOS ONE

Journal Requirements:

"This research was supported supported by the INVEST Research Flagship Centre/Research Council of Finland (decision number: 345546)."

Reviewers' comments:

Reviewer's Responses to Questions

**Comments to the Author**

1. Is the manuscript technically sound, and do the data support the conclusions?

Reviewer #1: Yes

Reviewer #2: Yes

Reviewer #3: Yes

Reviewer #4: Yes

2. Has the statistical analysis been performed appropriately and rigorously? 

Reviewer #1: Yes

Reviewer #2: Yes

Reviewer #3: Yes

Reviewer #4: Yes

3. Have the authors made all data underlying the findings in their manuscript fully available?

Reviewer #1: Yes

Reviewer #2: Yes

Reviewer #3: Yes

Reviewer #4: Yes

4. Is the manuscript presented in an intelligible fashion and written in standard English?

Reviewer #1: Yes

Reviewer #2: Yes

Reviewer #3: Yes

Reviewer #4: Yes

5. Review Comments to the Author

Reviewer #1: what the criteria of low or medium human development index (HDI) line 150

line 158 Coordination disorders are characterized by impaired fine and gross motor coordination development no surgical or accidents matters

line 200 25(OH)D vitamin D or 1, 25 Di (OH) D

vitamin D have role in immunity do you have any results about immunity or disease resistance

Reviewer #2: Prenatal vitamin D levels of immigrant mothers and language, scholastic, coordination and mixed developmental disorders in offspring

Vitamin D deficiency causes health issues not only in children as well in young and old age livings both in human and animals. Prenatal vitamin D levels of immigrant mothers and language, scholastic, coordination and mixed developmental disorders in offspring is an interesting study with respect to immigrant population in Finland. I will suggest few changes in the manuscript before publication. These changes are following:

On line # 84: write full spellings of ADHD as used first time in the text.

On line # 229: Can authors add data of % age of different countries rather than immigrant’s continents having vit D deficiency used in the present study samples?

was there any deficiency of Ca, P etc in immigrants mothers related with the deficiency of vit D?

is there any relationship between the year of migration of immigrant’s mothers, present study and Vit D deficiency?

Need to improve table #02 for readers.

Reviewer #3: I do not like the title of the paper. It needs to be made more precise. Otherwise, the paper is a very good attempt. Methodology is sound. Discussion is extremely comprehensive. Methodology is well detailed.

Reviewer #4: Congrats to the authors on their hard work. It is an original study—the introduction, discussion, and conclusion were well-written and robust statistical analysis performed. I would advise the author to consider enhancing the visual cues by adding more graphics, not just tables.

6. PLOS authors have the option to publish the peer review history of their article (what does this mean?). If published, this will include your full peer review and any attached files.

Reviewer #1: **Yes: **Hamada A. Ahmed

Reviewer #2: No

Reviewer #3: No

Reviewer #4: No

---

## [Author Response · Author response to Decision Letter 0]

30 Jan 2024

Thank you for providing us with some valuable feedback and comments. We have now revised the manuscript, see below for detailed responses and edits. 

A) Editorial comments

Response: we have checked the formatting requirements and modified the manuscript accordingly.

Response: the grant information has been checked and corrected accordingly (Academy of Finland and Research Council of Finland are the same organization, this has been clarified now). 

Thank you for stating the following financial disclosure: 

"This research was supported supported by the INVEST Research Flagship Centre/Research Council of Finland (decision number: 345546)."

Response: the role of the funder has been added to the cover letter.

Response: the legal restrictions that prohibit data sharing have been precised in the cover letter.

Response: We have added S3 Table with the relevant supporting data and removed the phrase “data not shown”.

Response: The reference list has been checked and is correct. No changes were made to the references.

B) Reviewers' comments

Reviewer #1:

1) what the criteria of low or medium human development index (HDI) line 150

Response: Thank you for pointing this out. We have added information on the HDI categorization to the manuscript (page 6, lines 173-177). The reference provided (ref 14) also offers extensive information on the categorization.

2) line 158 Coordination disorders are characterized by impaired fine and gross motor coordination development no surgical or accidents matters

Response: We have added this precision to page 6, line 181.

3) line 200 25(OH)D vitamin D or 1, 25 Di (OH) D

Response: This line was revised accordingly (line 228).

4) vitamin D have role in immunity do you have any results about immunity or disease resistance

Response: Thank you for the suggestion. Unfortunately, we do not have any data on immunological markers for this sample.

Reviewer #2: Prenatal vitamin D levels of immigrant mothers and language, scholastic, coordination and mixed developmental disorders in offspring

Vitamin D deficiency causes health issues not only in children as well in young and old age livings both in human and animals. Prenatal vitamin D levels of immigrant mothers and language, scholastic, coordination and mixed developmental disorders in offspring is an interesting study with respect to immigrant population in Finland. I will suggest few changes in the manuscript before publication. These changes are following:

5) On line # 84: write full spellings of ADHD as used first time in the text.

Response: “attention deficit hyperactivity disorder (ADHD)” has been added to the text, line 103.

6) On line # 229: Can authors add data of % age of different countries rather than immigrant’s continents having vit D deficiency used in the present study samples?

Response: We did not fully understand this comment. What is meant by “Age of different countries?” However, the S2 Appendix provides a list of all the included immigrant mothers’ birth countries.

7) was there any deficiency of Ca, P etc in immigrants mothers related with the deficiency of vit D?

Response: Thank you for the question. Unfortunately, nor calcium or phosphate were measured from the study subjects’ blood samples.

8) is there any relationship between the year of migration of immigrant’s mothers, present study and Vit D deficiency?

Response: Thank you for the excellent question. We checked this: the time from immigration date to blood draw was not associated with the vitamin D levels among immigrant mothers. We added this to the results, page 10, lines 284-288.

9) Need to improve table #02 for readers.

Response: Thank you for the suggestion. The table has been modified a little bit to be more reader-friendly, but the journal’s requirements prohibit the use of e.g., extra spacing, so the changes were minor. Please suggest more specific revisions to the table if needed.

10) Reviewer #3: I do not like the title of the paper. It needs to be made more precise. Otherwise, the paper is a very good attempt. Methodology is sound. Discussion is extremely comprehensive. Methodology is well detailed. 

Response: Thank you for the positive feedback. We changed the title to “Vitamin D levels of pregnant immigrant women and developmental disorders of language, learning and coordination in offspring” We are also open to other suggestions for the title.

11) Reviewer #4: Congrats to the authors on their hard work. It is an original study—the introduction, discussion, and conclusion were well-written and robust statistical analysis performed. I would advise the author to consider enhancing the visual cues by adding more graphics, not just tables.

Response: Thank you for the positive feedback! We have added a figure on the vitamin D level changes before and after the food fortification and recommendation change in Finland in 2003 (Fig 1).

---

## [Decision Letter · Decision Letter 1]

19 Feb 2024

Vitamin D levels of pregnant immigrant women and developmental disorders of language, learning and coordination in offspring

PONE-D-23-33791R1

Dear Dr. Arrhenius,

We’re pleased to inform you that your manuscript has been judged scientifically suitable for publication and will be formally accepted for publication once it meets all outstanding technical requirements.

Kind regards,

Ramune Jacobsen

Academic Editor

PLOS ONE

Reviewers' comments:

Reviewer's Responses to Questions

**Comments to the Author**

1. If the authors have adequately addressed your comments raised in a previous round of review and you feel that this manuscript is now acceptable for publication, you may indicate that here to bypass the “Comments to the Author” section, enter your conflict of interest statement in the “Confidential to Editor” section, and submit your "Accept" recommendation.

Reviewer #1: All comments have been addressed

Reviewer #2: All comments have been addressed

Reviewer #3: All comments have been addressed

Reviewer #4: All comments have been addressed

2. Is the manuscript technically sound, and do the data support the conclusions?

Reviewer #1: Yes

Reviewer #2: Yes

Reviewer #3: Yes

Reviewer #4: Yes

3. Has the statistical analysis been performed appropriately and rigorously? 

Reviewer #1: Yes

Reviewer #2: Yes

Reviewer #3: Yes

Reviewer #4: Yes

4. Have the authors made all data underlying the findings in their manuscript fully available?

Reviewer #1: Yes

Reviewer #2: Yes

Reviewer #3: Yes

Reviewer #4: Yes

5. Is the manuscript presented in an intelligible fashion and written in standard English?

Reviewer #1: Yes

Reviewer #2: Yes

Reviewer #3: Yes

Reviewer #4: Yes

6. Review Comments to the Author

Reviewer #1: The data provided as part of the manuscript or its supporting information. he statistical analysis been performed appropriately. Authors have adequately addressed my comments raised in a previous round of review and I feel that this manuscript is now acceptable for publication.

Reviewer #2: Authors addressed all the questions. Please, consider paper entitled as Vitamin D levels of pregnant immigrant women and developmental disorders of language, learning and coordination in offspring for publication.

Reviewer #3: Good work done. I think the authors have addressed all the queries which were raised by the reviewers.

Good work done. I think the authors have addressed all the queries which were raised by the reviewers.

Reviewer #4: The author addressed all the comments. They added the figures, changed the paper title and did all the reviewers suggested modifications.

7. PLOS authors have the option to publish the peer review history of their article (what does this mean?). If published, this will include your full peer review and any attached files.

Reviewer #1: **Yes: **Hamada A. Ahmed

Reviewer #2: No

Reviewer #3: No

Reviewer #4: No

---

## [Editor Report · Acceptance letter]

20 Feb 2024

PONE-D-23-33791R1 

PLOS ONE

Dear Dr. Arrhenius, 

I'm pleased to inform you that your manuscript has been deemed suitable for publication in PLOS ONE. Congratulations! Your manuscript is now being handed over to our production team.

Kind regards, 

on behalf of

Dr. Ramune Jacobsen 

Academic Editor

PLOS ONE